# Patient Self-Performed Point-of-Care Ultrasound: Using Communication Technologies to Empower Patient Self-Care

**DOI:** 10.3390/diagnostics12112884

**Published:** 2022-11-21

**Authors:** Andrew W. Kirkpatrick, Jessica L. McKee, Kyle Couperus, Christopher J. Colombo

**Affiliations:** 1TeleMentored Ultrasound Supported Medical Interventions (TMUSMI) Research Group, Calgary, AB T3H 3W8, Canada; 2Departments of Critical Care Medicine and Surgery, University of Calgary, Calgary, AB T2N 1N4, Canada; 3Ready Medic One (RMO) Research Group, Tacoma, WA 98431, USA; 4Department of Medicine, Uniformed Services University of Health Sciences Bethesda Maryland, Bethesda, MD 20814, USA

**Keywords:** point-of-care ultrasound, patient self-care, telementoring, informatics, community out-reach (Min. 5–Max. 8)

## Abstract

Point-of-Care ultrasound (POCUS) is an invaluable tool permitting the understanding of critical physiologic and anatomic details wherever and whenever a patient has a medical need. Thus the application of POCUS has dramatically expanded beyond hospitals to become a portable user-friendly technology in a variety of prehospital settings. Traditional thinking holds that a trained user is required to obtain images, greatly handicapping the scale of potential improvements in individual health assessments. However, as the interpretation of ultrasound images can be accomplished remotely by experts, the paradigm wherein experts guide novices to obtain meaningful images that facilitate remote care is being embraced worldwide. The ultimate extension of this concept is for experts to guide patients to image themselves, enabling secondary disease prevention, home-focused care, and self-empowerment of the individual to manage their own health. This paradigm of remotely telementored self-performed ultrasound (RTMSPUS) was first described for supporting health care on the International Space Station. The TeleMentored Ultrasound Supported Medical Interventions (TMUSMI) Research Group has been investigating the utility of this paradigm for terrestrial use. The technique has particular attractiveness in enabling surveillance of lung health during pandemic scenarios. However, the paradigm has tremendous potential to empower and support nearly any medical question poised in a conscious individual with internet connectivity able to follow the directions of a remote expert. Further studies and development are recommended in all areas of acute and chronic health care.

## 1. Introduction

Point-of-Care ultrasound (POCUS) is an invaluable tool permitting the understanding of critical physiologic and anatomic details wherever and whenever a patient has a medical need. Thus, the utilization of POCUS has dramatically expanded beyond hospitals to become a portable user-friendly technology in a variety of prehospital settings. Traditional thinking holds that a trained user is required to obtain images, greatly handicapping the potential improvement in individual health assessments. However, as the interpretation of ultrasound images can be accomplished remotely by experts, the paradigm wherein experts guide novices to obtain meaningful images that facilitate remote care is also being embraced worldwide. The ultimate extension of this concept is for experts to guide patients to image themselves, enabling secondary disease prevention, home-focused care, and self-empowerment of the individual to manage their own health. This paradigm of remotely telementored self-performed ultrasound (RTMSPU) was first described for supporting health care on the International Space Station. The TeleMentored Ultrasound Supported Medical Interventions (TMUSMI) Research Group thereafter investigated the utility of this paradigm for terrestrial use. The technique has particular attractiveness in enabling surveillance of lung health during Pandemic scenarios. However, the paradigm also has tremendous potential to empower and support nearly any medical question poised in a conscious individual with Internet connectivity able to follow the directions of a remote expert. Further, studying the interaction between remote experts and local individuals could be utilized to design artificial intelligence algorithms capable of guiding quality image acquisition without connectivity. Images could be sent asynchronously when limited connectivity is re-established and then interpreted, with recommendations from the expert sent back to the individual asynchronously. This approach could potentially further expand the utility of this technology to austere and disconnected environments.

## 2. Concepts

POCUS is an invaluable diagnostic tool for innumerable diagnostic, interventional, and educational uses. Thus the World Health Organization has long recognized ultrasound as one of the most important technologies the developing world requires and considers its access a minimal global standard [1,2]. Ultrasound technology and equipment have dramatically improved in terms of capability and portability while being ever more economical [3]. This allows powerful ultrasound equipment to be more readily available in many locations where there may not be a trained and experienced ultrasonographer. Fortunately, of all the imaging technologies, ultrasound imaging has long consisted of two linked but separate processes—image generation and image interpretation [4], and even in quaternary care hospitals, these processes are typically performed separately by ultrasound image generating technicians and interpreting radiologists. Thus, this enables the interpreting ultrasonographer to be in almost any location on the planet as long as reasonable two-way communication is permitted, such that a remote expert can guide the person holding the probe to obtain images that can be thus interpreted by the expert. When the person holding the probe is less experienced and is being mentored this concept is designated Remote Telementored Ultrasonography (or sonography) (RTMUS) [5,6,7]. When the novel person is being remotely mentored to generate images of themself, we have designated this as Remote Telementored Self-Performed Ultrasound (RTMSPUS) [8,9,10].

## 3. The Space Medicine Origins of Remote Telementored Self-Performed Ultrasound

Just as ultrasound is an ideal tool in resource limited settings on Earth [11], so it has been appreciated in space [12]. Thus, the only diagnostic imaging capability ever transported off the planet Earth into true space onboard the International Space Station, is diagnostic ultrasound [6,13]. Thus ultrasound has been incorporated as a backbone of space medicine protocols looking at various outside-the-box applications [14]. This paradigm has been accomplished despite the absence of a trained sonographer onboard the ISS through the use of informatics and remote guidance by remote ground-based experts using RTMUS [6,15,16]. In this paradigm, the ground-based expert(s) are responsible for guiding the astronaut in space to move the probe to generate real-time images that the terrestrial expert can see, correct, and evaluate. This experience has typically involved one astronaut imaging another, but it also introduced the RTMSPUS concept.

As there is extreme competition for every minute of time onboard the ISS, it was found to be more efficient to have an astronaut image themselves, as to have them image another crewmember which requires double the human resources. Thus, astronauts were guided and demonstrated the feasibility of RTMSPUS of the peritoneal cavity (Focused Assessment with Sonography for Trauma exam), echocardiography [17], a full urinary system examination of the retroperitoneum and pelvis [18], assessment of the jugular venous pressure (JVP) [19], and even panoramic ultrasound depictions of muscle mass [20], Self-scanning has particular attributes in weightlessness wherein Hamilton commented that it has the advantage of self-stabilization and no concern of the subject floating away from the operator in microgravity [17]. It was further noted that the sensitivity, specificity, and accuracy of the JVP in weightlessness were higher than those on Earth [19].

With the successful demonstration of RTMUS in general and its adoption within the core of Space Medicine practice and theory, the authors have long sought to translate these benefits for terrestrial care on Earth. The TeleMentored Ultrasound Supported Medical Interventions (TMUSMI) Research group thus initiated clinical work involving a fixed internet connection between a Quaternary care trauma center and a rural emergency room in the Rocky Mountains [21]. This project allowed a trauma surgeon in the receiving center to guide resuscitative ultrasound examinations by inexperienced providers at the small rural emergency. It facilitated remote diagnoses of an occult pneumothorax and cases of traumatic hemothorax including one direct to operating room resuscitation in an unstable patient [21]. Unfortunately, the paradigm was unsustainable due to the time delay required for the responding trauma surgeon to physically travel to a fixed teleultrasound base-station. The TMUSMI group has subsequently endeavored to refine RTMUS on completely mobile handheld platforms and to understand the human factors dynamics of RTMUS [7,22,23,24]. Through this research, it also became apparent that RTMUS is conceptually just a component of a richer telemedical interaction involving audio, visual, and often vital sign communication, as well as the visual information transmitted as ultrasound images.

## 4. Terrestrial Concepts of Remote Telementored Self-Performed Ultrasound (RTMSPUS)

If an individual is conscious and physically able to engage and participate in their own healthcare, then RTMUS offers unlimited capacity to dramatically increase the medical information transfer available to a remote expert attempting to guide care remotely. While caregivers have most frequently spoken to patients over the telephone to perform triage, the recent COVID pandemic dramatically increased the global utilization of videoconferencing as a provider-patient relationship [25,26]. However, one current limitation of telehealth is the inability to physically examine the patient remotely which leads to the risk of missed or improper diagnosis. Clinicians are thus appropriately concerned about diagnostic safety in telemedicine encounters, especially from a reduced or complete inability to perform a physical examination to collect information to formulate an accurate diagnosis [27,28].

RTMSPUS not only addresses this problem but improves upon the known limitations of the physical examination. Just as the physical examination dramatically improves the accuracy of a verbal history in confirming suspected diagnoses, so does a focused POCUS examination dramatically improve upon a physical examination [29,30,31]. Thus, the authors suggest that a self-performed ultrasound examination be considered whenever a patient is perceived to potentially benefit from an improved physical examination during a remote medical encounter. The existing database for self-performed ultrasound to date is summarized in Table 1. This table should be understood as representing current examples that will hopefully rapidly expand. The list of potential indications for RTMSPUS are as unlimited as the pathologies that afflict life on Earth.

## 5. Remote Telementored Self-Performed Ultrasound (RTMSPUS) as Means of Preserving Social Distancing in Pandemic Conditions

At the time of writing we continue to be in a global pandemic [32], and the world continues to suffer both directly from the diseases induced by the COVID-19 virus and from our societal responses to the challenges of containing a rapidly mutating very transmissible virus. Fortunately, the particular pathophysiology of COVID-19 pneumonia lends itself to screening, diagnosis, prognostication, and clinical follow up by POCUS. Early chest CT has been recommended early detection of suspected COVID-19 pneumonia with better sensitivity than polymerase chain reaction [33], but obtaining a home delivered CT scan is obvious impractical [34]. Fortunately LUS may have comparable results to chest CT with markedly reduced logistical challenges [35]. Thus, lung ultrasound has been recommended for increased use in all these roles [36,37,38,39,40,41]. Most of the mortality in COVID is due to respiratory failure, typically beginning ten days after exposure with the progression of infection from the nasopharynx to the lungs [42]. Point of care lung ultrasound is excellent for examining the pleural surfaces of the lungs [43], which is where lung swelling begins to accumulate when COVID infection progresses to COVID pneumonia. This is also when patients typically begin to deteriorate with sometimes marked life-threatening disease often despite notable symptoms, the so-called happy hypoxemia that may be disastrous [44,45]. In general, POCUS brings the caregiver performing ultrasound in intimate proximity to the infectious patient; thus, numerous procedures to try and protect the ultrasonographer have been recommended. Throughout the pandemic however, health care providers have often been disproportionately affected as they will likely be in future pandemics when the infective agents and transmission characteristics are most unknown. The World Health Organization estimated midway through the pandemic that between 80,000 and 180,000 health and care workers could have died from COVID-19 in the period between January 2020 to May 2021 [46]. While the effectiveness of societal social distancing remains controversial [47], there is no question that if a health care provider is never physically exposed to a potentially infective patient, then all will be safer. Further, if patients can be adequately assessed and followed in their own homes, this greatly reduces logistics, prevents community spread, and protects caregivers. Early on in the pandemic, physicians in an overwhelmed Northern Italian hospital pleaded with the world to adopt increased telemedical surveillance of self-isolating at risk patients to keep them out of the hospital as a means of preserving resources and saving lives [48]. TMUSMI strongly concurred with this sentiment and proposed that lung RTMSPUS could potentially be performed by willing patients to assess their own lung health remotely and to detect early changes that might prompt the need for earlier in-hospital assessment [9,34,49].

A limited experience suggests that health care professionals can self-image themselves satisfactorily (including for appendicitis) [50]. An emergency physician reported using home ultrasound upon themselves to confirm a COVID diagnosis and thereafter to follow herself to rule out other worrisome conditions, such as deep vein thrombosis, right heart strain from a massive pulmonary embolus, pericardial effusion, and lobar pneumonia [51]. Pivetta also reported a case of a nurse who had previous point-of-care ultrasound training on vascular access and bladder scanning who was “teleguided” by an expert operator to image her own lungs with a 12-zone protocol after she was diagnosed with COVID [52].

Although it is intuitive to take advantage of the existing skill sets of health care providers, it will be necessary to empower the general population during pandemic conditions. TMUSMI recently conducted a pilot trial in Edmonton, Canada wherein lay people without previous experience holding an ultrasound probe were guided by a mentoring expert in Calgary, 300 km distant [8]. It was subsequently demonstrated that all these healthy self-isolating non-medical volunteers were able to receive remote guidance to image themselves successfully. In this study as in all RTMSPUS situations, all interpretation of the LUS findings are the responsibility of the remote mentor. They were thus able to image their anterior, lateral, and bases of their backs with a 99.8% adequacy rate as assessed by blinded lung ultrasound reviewers (Figure 1).

Not unexpectantly, only two-thirds of this population could fully image their backs. While all could image their own low-back, only 70% could image their midback, and only 30% their upper backs for the posterior superior lung fields, in a group of whom 26% self-reported a shoulder injury and 30% upper body musculoskeletal problems [8]. Although anticipated, there was not an obvious difference between sexes. In the case of screening for COVID, this is not a practical limitation as the COVID hot spots at the bases of the lungs could be imaged in 96% of cases [8]. This fact illustrates a factor that does not typically arise in the standard paradigm of a technician performing an ultrasound on a patient, namely the physical ability of a subject to reach their own body parts with an ultrasound probe. Others are considering these facts though, and Kimura and colleagues also recently published a retrospective analysis of lung ultrasound findings in acutely ill patients with COVID and determined that viewing only the antero-apex of the lungs has had findings in 62% of all patients and associated with death and need for critical care unit admission [42].

One interesting sign that was used to confirm that the true lung bases were being seen, was to identify the liver and lung “points”. In this paradigm described by the TMUSMI group, the image under the stationary ultrasound probe can be seen to alternate between a view of the contiguous visceral and parietal pleura and alternatively a view of the underlying liver or spleen depending on which side of the body (Figure 2, Appendix A), as the diaphragm and relative visceral positions alternates with respiration. This sign again 100% confirms that the lung bases are being imaged.

Resnikoff also recently reported a mixed cohort of 80 COVID confirmed, dyspneic, or cardiology patients who were able to self-image the anterior second intercostal space bilaterally with a 67% sensitivity, 94% specificity, and 88% accuracy for standard lung ultrasound landmarks, with the participants being guided only by a instruction sheet [53]. They also noted that those younger than 80 who used the internet daily were more likely to obtain diagnostic lung images, while being acutely dyspneic was associated with reduced success [53].

Thus, TMUSMI believes it would be possible for large populations of asymptomatic or paucisymptomatic patients at potential but a low risk of deterioration to be screened as frequently as deemed appropriate in their own households. Sanitized lightweight telemedical smart phone supported ultrasound probes can be quickly couriered or even drone delivered to completely obviate human to human contact (Figure 3a,b) [54,55]. In the group of previously asymptomatic patients at risk for COVID, one subject later caught COVID, with his initial pre-infectious study serving as a baseline for repeated examinations by the same remote mentor providing longitudinal follow-up to detect deterioration at the earliest possible time [10].

## 6. Self-Performed Home Obstetrical Ultrasound

In 2011, TMUSMI reported early experiences with the transnational remote mentoring of maternal wellness examinations [7]. There have since been remarkable developments including the development of self-scanning algorithms that allow pregnant women to image their own fetus’s guided by an app to guide the mothers through self-scanning with the results viewed remotely in either delayed or real-time. Hader reported that among 100 women, there was 95.3% success for detecting fetal heart activity, 88.3% for body movements, and 92% for normal amniotic fluid volume, although only 23% success in detecting fetal breathing. They concluded the system was a potential solution for remote sonographic fetal assessment although further study was required [56].

## 7. The Potential of Remotely Mentored Teleultrasound for Acute Self Resuscitation

With the dramatically increased availability of ultrasound devices that are now connected and powered by smart phones, it is likely any severely injured victim who is not unconscious, will be able to reach out for potential online remote help, which might include the possibility of remotely directed self-performed ultrasound assessments to assist in triage efforts, or even remotely guided self-care in extreme circumstances when no other options exist. This topic was recently reviewed by TMUSMI in the publication entitled; “Empowering catastrophic far-forward self-care: Nobody should die alone without trying” [57]. Just as it would be strange for a trauma surgeon to function in a trauma resuscitation bay without ultrasound, it may become the norm for a remote rescuer to supplement the basic “how are you” questions in an emergent medical interaction with a progressive ultrasound vital signs check.

## 8. The Potential of Remotely Mentored Teleultrasound for Self-Care of Chronic Conditions

While TMUSMI has been focused on acute health events, managing chronic health problems of an aging population challenge is a profound global challenge, and one that may bankrupt publicly funded health care systems. Thus, any solution that can keep the aging safely at home in good health, rather than requiring in hospital assessment or institutionalization will not only dramatically reduce costs but hopefully increase the quality of life. Smart homes are one form of gerontechnological innovation, that designates the augmentation of a residence with sensors and devices/actuators integrated into the residence’s infrastructure. While occupants have been satisfied with their residences, they have not been proven to reduce emergency department visits or hospital admission rates [58]. TMUSMI proposes that adding a remote ultrasound capability to the typical smart home infrastructure may provide the ability to improve home care in almost limitless ways. Low-lying fruit already being include remote self-monitoring of cardiac conditions such as congestive heart failure [53]. By potentially using RTMSPUS innumerable conditions such as pneumothorax, pulmonary edema, pleural effusion, aspiration pneumonia, and pneumonia might be remotely diagnosed remotely [59]. As ultrasound is the most versatile imaging technology without side effects; thus it is presumable that almost any medical, nursing, and allied specialty will find some aspect of RTMSPUS to incorporate into their improved telehealth interactions.

## 9. Future Directions and Conclusions

The COVID pandemic redefined much of Global Societies communicate, including catalyzing the acceptance and demand for remote services including acute and chronic health needs. While remote access is potentially safer and more convenient, the risk of missing important findings due to the lack of in-person assessment is ever present. Enabling patients to examine themselves with smart phone powered and supported ultrasound brings the care giver back to the patient’s presence and improves the physical examination, providing immediate answers that can be electronically documented for future analysis or comparison. TMUSMI and others have shown that almost any technique is possible. The potential for impacting triage, diagnosis and treatment decisions extends in the present to high connectivity environments, and in the immediate future to poorly connected environments. The challenge is to make these techniques logistically sustainable, to understand the human factors required to provide optimal mentoring of this technique, and to improve artificial intelligence and machine learning to the point of extending this resource to disconnected environments.

## Figures and Tables

**Figure 1 diagnostics-12-02884-f001:**
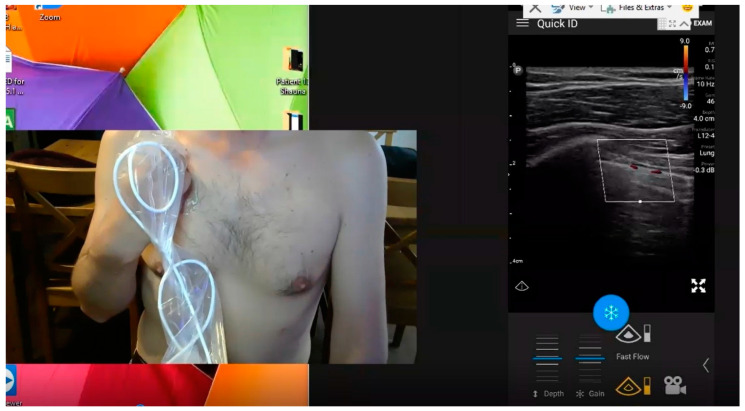
Example of Ultrasound Naïve Volunteer being guided to image their own chest. Figure Legend: Screenshot of the Mentors computer in Calgary viewing the completely novice self-isolating volunteer image their upper right anterior lung field depicting the visceral and parietal pleural interface with movement emphasized with Color Power-Doppler the “Power-Slide” Sign (seen in Appendix A). Note: Lung ultrasound is a dynamic science much better appreciated with real time imaging. The videorecording of the entire mentored Lung examination ca be viewed in Appendix A.

**Figure 2 diagnostics-12-02884-f002:**
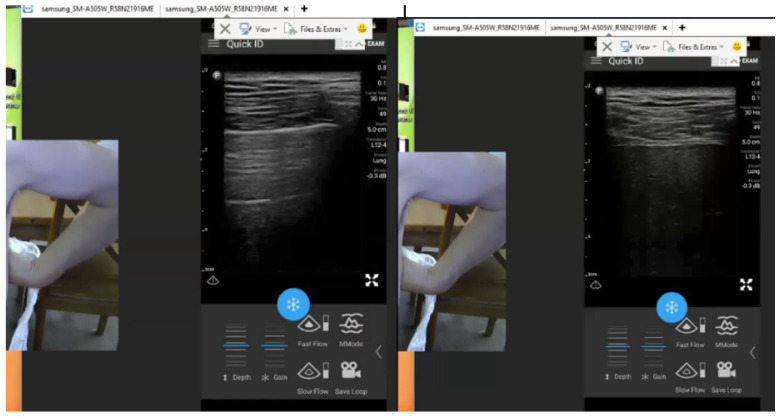
Example of the “Spleen Point” Sign confirming imaging of the Lung Base. Legend: When the ultrasound probe is correctly positioned at the bung base, the image will review the visceral-parietal interface (left) alternating with the parenchyma of the spleen on the left or liver on the right with no probe movement due to the respiratory movement of the diaphragm. Note: Lung ultrasound is a dynamic science much better appreciated with real time imaging. The videorecording of the entire mentored Lung examination ca be viewed in Appendix A.

**Figure 3 diagnostics-12-02884-f003:**
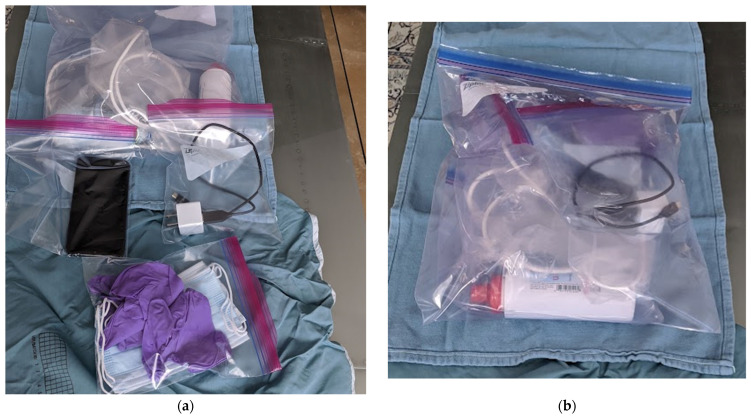
(**a**) Sanitized Home Ultrasound Delivery Package. Legendz: Sanitized package for home-delivery to patients containing a Philips Lumify Ultrasound probe and dedicated smart phone to support the ultrasound, gloves, masks, and ultrasound jelly. (**b**) Sanitized Home Ultrasound Delivery Package. Figure Legend: Sanitized package ready for home-delivery to self-isolating patients.

**Table 1 diagnostics-12-02884-t001:** Indications for Remote TeleMentored Self-Performed Ultrasound (RTMSPUS).

Indication	Existing or Theoretical Status
**Space Medicine**	RTMSPUS has been performed since the early provision of an ultrasound machine onboard the International Space Station for a multitude of indications
**COVID Lung examinations**	Reported as case reports and case series during the current COVID 19 pandemic
**Maternal Wellness Examination**	Reported as case series emphasizing practicality
**Heroic Self Preservation**	If a severely injured but isolated casualty is conscious then self-diagnosing and providing mentored self-care may be the only option for saving their own life reported as theoretical concepts
**Appendicitis**	Reported in case report format

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
