# Peer review of "Patient Self-Performed Point-of-Care Ultrasound: Using Communication Technologies to Empower Patient Self-Care"

_diagnostics, 2022, doi:10.3390/diagnostics12112884_

Round 1

Reviewer 1 Report

The review by Andrew W Kirkpatric et al was generally well designed and well organized. There are some excellent studies relating to lung ultrasound examination of COVID19 missed in the review. This reviewer would like to suggest for publication after including these literatures.

Author Response

“The review by Andrew W Kirkpatrick et al was generally well designed and well organized.  There are some excellent studies relating to lung ultrasound examination of COVID19 missed in the review. This reviewer would like to suggest for publication after including this literature.”

Reply

We thank the Reviewer very much for their time and efforts.

We have also attempted to succinctly mention the many uses and benefits to patients in the discussion that previously read;

Fortunately, the particular pathophysiology of COVID-19 pneumonia lends itself to screening, diagnosis, prognostication, and clinical follow up by POCUS. Thus, lung ultrasound has been recommended for increased use in all these roles(1-3).  Most of the mortality in COVID is due to respiratory failure, typically beginning ten days after exposure with the progression of infection from the nasopharynx to the lungs(4).  Point of care lung ultrasound is excellent for examining the pleural surfaces of the lungs(5), which is where lung swelling begins to accumulate when COVID infection progresses to COVID pneumonia.  This is also when patients typically begin to deteriorate with sometimes marked life-threatening disease often despite notable symptoms, the so-called happy hypoxemia that may be disastrous(6, 7). 

We have thus added some exceptional studies regarding COVID 19 and lung ultrasound to the manuscript as follows;

Toraskar K, Zore RR, Gupta GA, Gondse B, Pundpal G, Kadam S, et al. Utility and diagnostic test properties of pulmonary and cardiovascular point of care ultra-sonography (POCUS) in COVID-19 patients admitted to critical care unit. Eur J Radiol Open. 2022;9:100451.

Skaarup SH, Aagaard R, Ovesen SH, Weile J, Kirkegaard H, Espersen C, et al. Focused lung ultrasound to predict respiratory failure in patients with symptoms of COVID-19: a multicentre prospective cohort study. ERJ Open Res. 2022;8(4).

Fang Y, Zhang H, Xie J, Lin M, Ying L, Pang P, et al. Sensitivity of Chest CT for COVID-19: Comparison to RT-PCR. Radiology. 2020:200432.

Kirkpatrick AW, McKee JL. Re: "Proposal for International Standardization of the Use of Lung Ultrasound for Patients With COVID-19: A Simple, Quantitative, Reproducible Method"-Could Telementoring of Lung Ultrasound Reduce Health Care Provider Risks, Especially for Paucisymptomatic Home-Isolating Patients? J Ultrasound Med. 2021;40(1):211-2.

Peng QY, Wang XT, Zhang LN, Chinese Critical Care Ultrasound Study G. Findings of lung ultrasonography of novel corona virus pneumonia during the 2019-2020 epidemic. Intensive Care Med. 2020.

Ma IWY, Hussain A, Wagner M, Walker B, Chee A, Arishenkoff S, et al. Canadian Internal Medicine Ultrasound (CIMUS) Expert Consensus Statement on the Use of Lung Ultrasound for the Assessment of Medical Inpatients With Known or Suspected Coronavirus Disease 2019. J Ultrasound Med. 2020.

Such that the new paragraph has a more detailed bibliography in relation to lung ultrasound and COVID-19 that reads;

“At the time of writing we continue to be in a global pandemic(8), and the world continues to suffer both directly from the diseases induced by the COVID-19 virus and from our societal responses to the challenges of containing a rapidly mutating very transmissible virus.  Fortunately, the particular pathophysiology of COVID-19 pneumonia lends itself to screening, diagnosis, prognostication, and clinical follow up by POCUS.  Early chest CT has been recommended early detection of suspected COVID-19 pneumonia with better sensitivity than polymerase chain reaction(9), but obtaining a home delivered CT scan is obvious impractical(10).  Fortunately LUS may have comparable results to chest CT with markedly reduced logistical challenges (11).  Thus, lung ultrasound has been recommended for increased use in all these roles(1-3, 12-14).  Most of the mortality in COVID is due to respiratory failure, typically beginning ten days after exposure with the progression of infection from the nasopharynx to the lungs(4).  Point of care lung ultrasound is excellent for examining the pleural surfaces of the lungs(5), which is where lung swelling begins to accumulate when COVID infection progresses to COVID pneumonia.  This is also when patients typically begin to deteriorate with sometimes marked life-threatening disease often despite notable symptoms, the so-called happy hypoxemia that may be disastrous(6, 7).  In general, POCUS brings the caregiver performing ultrasound in intimate proximity to the infectious patient; thus, numerous procedures to try and protect the ultrasonographer have been recommended. Throughout the pandemic however, health care providers have often been disproportionately affected as they will likely be in future pandemics when the infective agents and transmission characteristics are most unknown.  The World Health Organization estimated midway through the pandemic that between 80 000 and 180 000 health and care workers could have died from COVID-19 in the period between January 2020 to May 2021(15).  While the effectiveness of societal social distancing remains controversial(16), there is no question that if a health care provider is never physically exposed to a potentially infective patient, then all will be safer.  Further, if patients can be adequately assessed and followed in their own homes, this greatly reduces logistics, prevents community spread, and protects caregivers.  Early on in the pandemic, physicians in an overwhelmed Northern Italian hospital pleaded with the world to adopt increased telemedical surveillance of self-isolating at risk patients to keep them out of the hospital as a means of preserving resources and saving lives(17).  TMUSMI strongly concurred with this sentiment and proposed that lung RTMSPUS could potentially be performed by willing patients to assess their own lung health remotely and to detect early changes that might prompt the need for earlier in-hospital assessment(18-20).”

We will also happy to consider any of the multitude of excelent manuscripts concerning COVID-19 and lung ultrasoud if suggested by the Diagnostics Editotrial Staff or the REviwers.

Reviewer 2 Report

This is a well-written and avant garde manuscript describing the history of an emerging application of diagnostic ultrasound wherein the image acquisition and image interpretation are performed in physically separate locations: either by a clinician (RTMUS) or by the patient themselves RTMSPUS).  The authors draw a historical line beginning from work on the International Space Station to pilot work they themselves have conducted to connect a quarternary academic medical center with a rural hospital ER.  The author's pilot project identified several pain points that the authors are currently attempting to address, including the need for the image interpreting provider to have access to images/clips on a mobile platform and the importance of embedding diagnostic POCUS into a multi-modal telemedicine interaction that also includes things like audio, visual, and vital sign information.  The authors then review the literature on the topic of self-scanning with remote performance of both image acquisition mentoring and image interpretation.  The authors also address obvious potential barriers to implementation of patient self-scanning, including describing means by which patients may be able to receive just-in-time home ultrasound scanning kits to permit the RTMSPUS workflow.

Overall, I think the article is well-written and provocative in a good way.  Although there remain many logistical problems to implementing RTMSPUS on a global scale (most notably the difficulty of getting ultrasound probes + other ultrasound-relevant equipment to patients when they need them without damage/loss of equipment and enormous transportation costs, etc.), I think the authors do enough acknowledge these problems and explain why it is plausible that these obstacles could one day be surmounted.

Try as I might, I cannot think of anything specific I would want the authors to change.  I think this is worthy of publication as is.

Author Response

“Try as I might, I cannot think of anything specific I would want the authors to change. I think this is worthy of publication as is.”

Reply

We thank the Reviewer very much for their time and efforts.

I will of course be happy to make any other requested changes to the manuscript.